# TransformMix: Learning Transformation and Mixing Strategies from Data

## Abstract

Data augmentation improves the generalization power of deep learning models by synthesizing more training samples. Sample-mixing is a popular data augmentation approach that creates additional data by combining existing samples. Recent sample-mixing methods, like Mixup and Cutmix, adopt simple mixing operations to blend multiple inputs. Although such a heuristic approach shows certain performance gains in some computer vision tasks, it mixes the images blindly and does not adapt to different datasets automatically. A mixing strategy that is effective for a particular dataset does not often generalize well to other datasets. If not properly configured, the methods may create misleading mixed images, which jeopardize the effectiveness of sample-mixing augmentations. In this work, we propose an automated approach, *TransformMix*, to learn better transformation and mixing augmentation strategies from data. In particular, *TransformMix* applies learned transformations and mixing masks to create compelling mixed images that contain correct and important information for the target tasks. We demonstrate the effectiveness of *TransformMix* on multiple datasets in transfer learning, classification, object detection, and knowledge distillation settings. Experimental results show that our method achieves better performance as well as efficiency when compared with strong sample-mixing baselines.

## 1 Introduction

Modern deep learning models achieve remarkable success in important computer vision tasks, like object classification (Krizhevsky et al., 2012; He et al., 2016b; Zagoruyko & Komodakis, 2016) and semantic segmentation (Ren et al., 2015; Bochkovskiy et al., 2020). Despite these reported successes, deep learning models can easily overfit when the training set is quantitatively deficient. To generalize deep learning models beyond finite training sets, data augmentation is a widely adopted approach that synthesizes additional samples to expand the training sets (Shorten & Khoshgoftaar, 2019). Conventional data augmentation applies pre-defined image processing functions, such as random cropping, flipping and color adjustment, to create additional views of the same data. To reduce the manual effort in searching for the appropriate augmentation configuration, automated data augmentation (AutoDA) methods have been proposed to search for the optimal augmentation policy for a dataset (Cubuk et al., 2019; Ho et al., 2019; Lim et al., 2019; Cheung & Yeung, 2021; 2022). Given the right choice of augmentation, these transformation-based techniques induce constructive inductive biases in the dataset, thereby improving the generalization power of machine learning models.

Sample-mixing is a different data augmentation approach that synthesizes additional samples by combining multiple images. Unlike conventional data augmentation, sample-mixing does not require the specification of domain-specific transformations, allowing it to be flexibly deployed to other data domains. The seminal work Mixup interpolates two training images with their one-hot-encoded label proportionally (Zhang et al., 2018), while CutMix randomly replaces a patch of an image with another image (Yun et al., 2019). Although these mixing strategies can bring slight improvements in some computer vision tasks (Bochkovskiy et al., 2020), images are combined without considering their content. Consequently, the mixed images may contain misleading training signals and undermine the model performance. Specifically, Mixup may lead to the manifold collision problem where an interpolated image sits on the real data manifold (Guo et al., 2019);

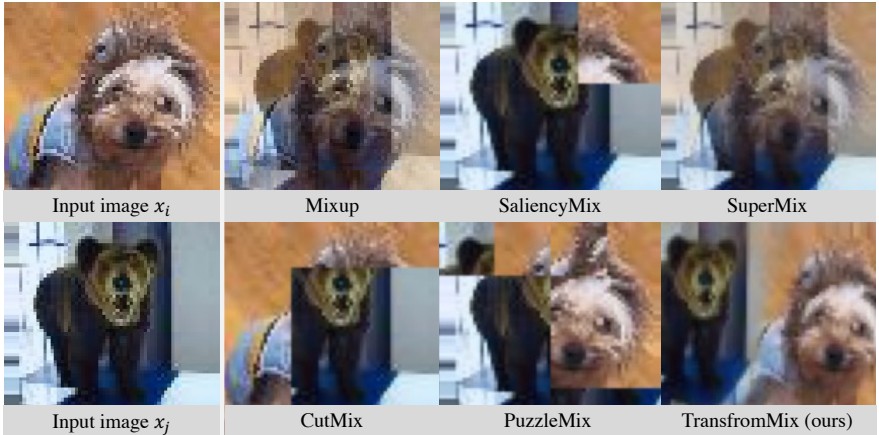

Figure 1: Visual comparison of different sample-mixing methods on a dog and bear image. Our method better preserves the important region of the input images.

the random replacement in CutMix may remove important regions that are crucial in identifying an object instance in classification tasks. To this end, recent works apply additional saliency constraints to avoid the crucial information being removed during the mixing process (Kim et al., 2020; Dabouei et al., 2021; Kim et al., 2021; Liu et al., 2021; Uddin et al., 2021).

Inspired by recent AutoDA advancements and saliency-aware mixing methods, we develop an automated method, named *TransformMix*, that learns a better mixing strategy from a dataset with two criteria: (1) the mixing strategy should create mixed outputs that maximally preserve the visual saliency of the input images; (2) the mixing strategy should be learned from the dataset automatically. For more practical usage in transfer learning, we also investigate whether a discovered mixing strategy can be transferred to create new augmented images on unseen datasets. At a high level, *TransformMix* exploits self-supervision signals from a pre-trained teacher network to learn a mixing module for predicting the transformations and mixing masks to create mixed images that preserve the salient information of the input images. To summarize, we make four major contributions.

- We propose *TransformMix*, which employs a novel mixing module that predicts the transformations and mixing masks to create more advantageous mixed images with maximal preservation of visual saliency.

- We introduce an automated framework that trains the mixing module on a given dataset efficiently.

- We demonstrate that our method improves state-of-the-art results on several popular classification benchmarks and achieves $4\times$ to $18\times$ speed-up compared to other saliency-aware sample-mixing baselines.

- We demonstrate that our method can be transferred to augment new unseen datasets. The transferred method provides non-trivial improvements over other sample-mixing methods. We also show the effectiveness of *TransformMix* on classification, object detection, and knowledge distillation tasks.

## 2 Related Works

**Automated Data Augmentation.** Conventional data augmentation applies label-preserving transformations to augment data. However, the specification and choice of data augmentation functions rely heavily on expert knowledge or repeated trials. To address this shortcoming, AutoDA learns the optimal policy to augment the target dataset (Cheung & Yeung, 2023). AutoAugment takes a reinforcement learning approach

to learn the probability and magnitude of applying different transformations to the target dataset (Cubuk et al., 2019). PBA (Ho et al., 2019) and RandAugment (Cubuk et al., 2020) study more efficient search methods to reduce the expensive search effort of AutoAugment. Inspired by AutoAugment, AdaAug (Cheung & Yeung, 2022) learns an adaptive augmentation strategy for each data instance, while MODALS (Cheung & Yeung, 2021) learns the optimal augmentation strategy to apply four transformations in the latent space, thereby allowing the method to be deployed to multiple modalities. Despite the successes reported by these AutoDA methods, they require domain knowledge in designing domain-specific label-preserving transformations. Moreover, they have not studied the optimal way to mix multiple images.

**MixUp.** MixUp (Zhang et al., 2018) uses the convex combination of input images as new augmented images: $x' = \lambda x_1 + (1 - \lambda)x_2$, where $x_1$ and $x_2$ are two training images, $\lambda$ is the mixing coefficient and $x'$ is the augmented image. Following Mixup, Manifold Mixup (Verma et al., 2019) applies Mixup to mix the latent representations, while CutMix (Yun et al., 2019) replaces a random patch of an image with another image. These techniques disregard the image content and may dilute or occlude the salient information, which is crucial for the target task. To preserves all saliency information, StackMix (Chen et al., 2022) concatenates two input images as the augmented data. Apart from studying new ways to mix the input images, other attempts have been made to prevent MixUp from creating out-of-distribution images. Specifically, Local MixUp (Baena et al., 2022) considers the locality of input samples and assigns lower loss weightings for distant samples; GAN-MixUp (Sohn et al., 2020) generates synthetic data that are close to the class boundaries and applies MixUp between the real and generated samples. On the other hand, $k$-MixUp (Greenewald et al., 2023) and Co-MixUp (Kim et al., 2021) extend the standard MixUp procedure to mix more than two inputs at a time.

NEW

**Saliency-aware Sample-mixing.** To mitigate the mentioned problem, several recent works attempted to create more advantageous mixed images by preserving the salient information in the mixed image. SaliencyMix (Uddin et al., 2021) and ResizeMix (Qin et al., 2020) detect the saliency information and prevent CutMix from replacing the image patch that contains rich information. SuperMix utilizes a teacher model to optimize some mixing masks applied to the input images: $x' = m_1 \odot x_1 + m_2 \odot x_2$, where $m_1$ and $m_2$ denote the mixing masks and $\odot$ denotes the elementwise multiplication. In AutoMix (Liu et al., 2021), the mixing masks are computed by a mix block based on the image features. However, consider two input images having the salient information at the same pixel location, for example, the face of a dog in input image $x_i$ and the face of a bear in input image $x_j$ as illustrated in Figure 1, optimizing a mixing mask alone cannot prevent the two faces being blended in the mixed results. Hence, PuzzleMix (Kim et al., 2020) optimizes the mixing masks together with transport plans $\Pi_1$ and $\Pi_2$ that specify the mass to be transported between different pixel locations: $x' = m_1 \odot \Pi_1 x_1 + m_2 \odot \Pi_2 x_2$. Although the method avoids the overlapping of the salient regions to a certain degree, the method splits the input images into different blocks and shifts their locations. This creates puzzle-like artifacts in the resulting image, violating the natural image prior (see Figure 1).

*TransformMix* aims to learn the transformation and mixing strategies that generate more advantageous mixed data automatically. Learning such strategies poses two major challenges. First, a mixing strategy needs to decide the output of every pixel location, which is harder than learning an augmentation policy from a set of 10 to 20 augmentation functions in previous AutoDA works. Therefore, formulating the transformation and mixing strategy and optimizing it efficiently is a challenging problem. Second, previous works create mixed input with unnatural mixing boundaries or require manual specification of additional constraints to create more realistic outputs. Designing an automated approach to generate more natural blends of images is non-trivial. Hence, developing a transferable mixing strategy like *TransformMix* to reduce the computation efforts on new datasets is a favorable solution for data practitioners.

## 3 Methodology

*TransformMix* first learns a mixing strategy from a dataset under the supervision of a pre-trained teacher model and then creates mixed data to train some end task networks. We formulate the mixing strategy by predicting some transformations and mixing masks applied to the input images. Given two distinct instances $(x_i, y_i)$ and $(x_j, y_j)$, where $x_i \in \mathbb{R}^{C \times W \times H}$ is the $i$-th training sample with class label $y_i$, $C$ channels and $W \times H$ dimension, *TransformMix* aims to find the effective transformations $\phi_i, \phi_j$ and mixing masks

$m_i, m_j \in [0, 1]^{W \times H}$ applied to $x_i$ and $x_j$ for creating better mixed images $x'$ according to:

$$x' = m_i \odot \phi_i(x_i) + m_j \odot \phi_j(x_j), \quad y' = \lambda y_i + (1 - \lambda)y_j, \tag{1}$$

where $\lambda$ is a mixing coefficient for mixing $x_i$ and $x_j$. Under this formulation, the transformations help to separate the salient regions even if they completely overlap in the input images, while the mixing masks help to reveal the more important regions of the candidate images with respect to the target task. To better preserve visual saliency, *TransformMix* utilizes the class activation maps (CAMs) of the input images when predicting the mixing strategy. The intermediate visual illustrations of the input images, saliency maps, transformations and predicted mixing masks are presented in Figure 2. In the following section, we explain the details of each component and then the training framework of *TransformMix*.

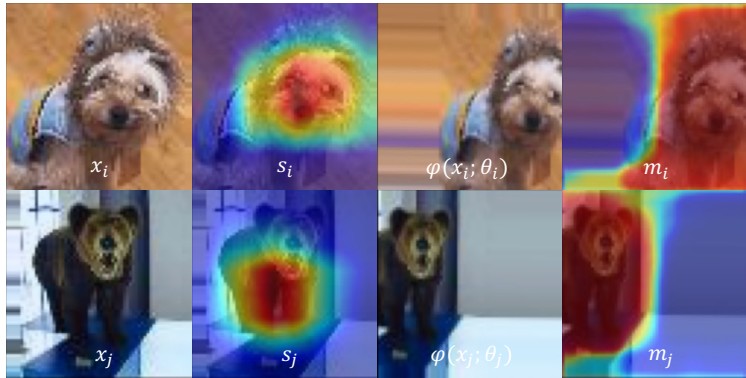

Figure 2: Illustrations of the intermediate results during mixing. From left to right, the columns show the visualizations of the input images, CAMs, transformed images and predicted masks.

### 3.1 TransformMix

*TransformMix* comprises three components: a pre-trained teacher model, a saliency detector and a mixing module. *TransformMix* first trains the mixing module under the supervision of the teacher model. Once the mixing module is trained, it is used to create mixed images to train a new task network. The overview of *TransformMix* is shown in Figure 3.

**Saliency Detector.** It was previously suggested that considering the image saliency during mixing can improve the quality of mixed images (Kim et al., 2020; Uddin et al., 2021). In our method, we adopt a neural network approach, which takes visual saliency as input features to make a prediction of a mixing strategy. We propose to condition on the visual saliency rather than the image or latent image representation because visual saliency is a generic measure applicable to all images, whereas the image or latent image representation is mostly restricted to a dataset. Therefore, learning from visual saliency facilitates transferring the trained mixing module to unseen data.

Typically, a saliency detector generates a heatmap $s_i \in [0, 1]^{W \times H}$ that highlights the important regions of $x_i$. In practice, we can use any saliency detection algorithms or explainable AI methods to extract salient information from input images. As *TransformMix* employs a pre-trained teacher network in the later training stage, we exploit the readily available pre-trained weights and use CAMs (Zhou et al., 2016) to estimate the salient regions of an image. In essence, CAMs are the summation of global averaged features weighted by the weights learned by the classification layer. The pixel location that is more important to the classification task will be assigned a larger value in the heatmap.

**Spatial Transformer Network.** Unlike PuzzleMix (Kim et al., 2020) which divides the input images into multiple blocks and re-organizes them to avoid overlapping salient regions, *TransformMix* encompasses a more comprehensive set of transformations and creates mixed outputs complying with the natural image prior. As a learnable transformation method, Spatial Transformer Network (STN) is a convolutional neural network module that spatially transforms an image by predicting the appropriate transformation for each

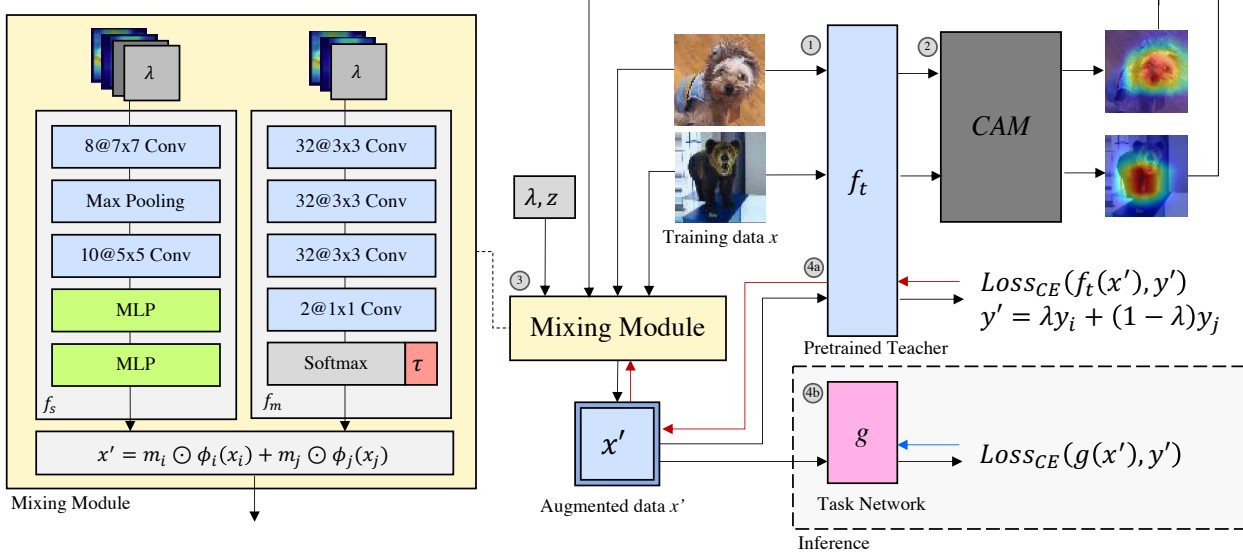

Figure 3: Overview of *TransformMix*. The black arrows indicate the forward pass; the red arrows indicate the gradient flow when training the spatial transformation network $f_s$ and mask prediction network $f_m$; the blue arrow indicates the gradient flow when training the task network $g$. **Step 1&2**: the CAMs of the input images $(x_i, x_j)$ are extracted from the pre-trained teacher $f_t$. **Step 3**: the CAMs, input images and sampled mixing coefficient $\lambda$ are supplied to the mixing module to compute the transformations $(\phi_i, \phi_j)$ and mixing masks $(m_i, m_j)$ to create the mixed image $x'$, which is used in **step 4a**: training the mixing module, or **step 4b**: training the the task network.

data instance (Jaderberg et al., 2015). Different from the previous works that predict the transformation based on a single image, we propose a novel approach using an STN to predict 6 affine parameters for each of the two input images based on their CAMs, a mixing coefficient $\lambda$ and a sampled noise $z$. Specifically, the 6 affine parameters include a $2 \times 2$ sub-matrix that defines the linear transformation and 2 parameters that define the horizontal and vertical translations. The mixing coefficient $\lambda$ allows more emphasis on different   FIX proportions of mixing and is sampled from the Beta distribution with parameter $\alpha$. Sampled from the Normal distribution, $z$ adds certain diversity when transforming the images. The mixing coefficient and sampled noise are resized and appended to the CAMs as additional channels. We use $f_s : \mathbb{R}^{4 \times W \times H} \to \mathbb{R}^{2 \times 6}$ to denote the STN and $\theta$ to denote the predicted affine parameter, which is given by:

$$\theta_i, \theta_j = f_s(s_i, s_j, \lambda, z), \lambda \sim \text{Beta}(\alpha, \alpha), z \sim N(0, I) \tag{2}$$

Using the affine parameters predicted by $f_s$, the transformations are then performed on the input images accordingly to avoid overlapping of salient regions. Although recent AutoDA methods can search in a larger set of transformations, like color adjustments and other non-differentiable operations, the spatial transformations, including scaling, cropping, rotations, as well as non-rigid deformations, are sufficient to tackle the saliency overlapping issue. Therefore, we opt for the differentiable STN approach, which can be optimized efficiently using gradient-based methods, instead of costly AutoDA search methods, to characterize $\phi$ in equation 1.

**Mask Prediction Network.** The mask prediction network $f_m$ receives the transformed CAMs and mixing coefficient $\lambda$ to predict the mixing masks $(m_i, m_j)$. We implement $f_m$ as a spatial-preserving convolutional neural network, i.e., $f_m : \mathbb{R}^{3 \times W \times H} \to [0, 1]^{2 \times W \times H}$. The softmax function is applied to the output layer to ensure that the mixing masks sum to 1 at each pixel location $(w, h)$. In addition, a learnable temperature parameter $\tau$ is introduced to control the smoothness of the mixing boundary. Specifically, a lower temperature value will result in a sharper blending boundary (see Figure 4). With $o$ denoting the hidden feature before the softmax layer and $\varphi(\cdot; \theta)$ denoting the affine transformation with parameter $\theta$, the mixing masks are

computed as:

$$m_i, m_j = f_m(\varphi(s_i; \theta_i), \varphi(s_j; \theta_j), \lambda), \quad m_i^{(w,h)} = \frac{\exp(o_i^{(w,h)}/\tau)}{\sum_{k \in \{i,j\}} \exp(o_k^{(w,h)}/\tau)} \tag{3}$$

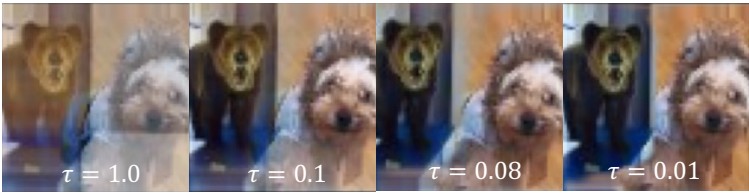

Figure 4: Illustrations of the effect when using different temperature values $\tau$. The third image is the mixed result with learned $\tau = 0.08$.

## 3.2 Training

*TransformMix* employs a two-stage training process. At the search stage, the mixing module is trained to learn the transformation and mixing strategy from a dataset. At inference time, the learned mixing module will generate new mixed images for training a new classifier on the dataset.

In AutoDA, augmentation policies are optimized with the goal to maximize the validation performance on a hold-out dataset directly. The process requires the training of multiple child models, which is computationally expensive to achieve. Specifically, AutoAugment spends over 5,000 GPU hours learning the augmentation policy on CIFAR-10 (Cubuk et al., 2019). To this end, we adopt a surrogate objective that correlates to the effectiveness of mixed training images. Inspired by SuperMix (Dabouei et al., 2021), we utilize a pre-trained teacher model $f_t$ to guide the learning of the mixing module. Specifically, we update the mixing module to minimize the classification loss on the mixed input $x'$ with label $y'$ in equation 4. With the supervision signals given by the pre-trained network, the mixing module $(f_s, f_m)$ learns to construct mixed images such that $f_t$ can uncover the constituting objects in $x'$. This encourages the mixing module to fit more important information in the mixed results, avoiding the salient information being diluted implicitly.

$$x' = m_i \odot \varphi(x_i; \theta_i) + m_j \odot \varphi(x_j; \theta_j), \;\; y' = \lambda y_i + (1 - \lambda)y_j \tag{4}$$

At inference time, the trained mixing module creates mixed images to train a new classifier following the standard model training process. In the previous sections, we explained the details of *TransformMix* to mix two input images. In practice, *TransformMix* can be generalized to mix more images by replacing the Beta distribution with Dirichlet distribution and modifying the input and output sizes of $f_s$ and $f_m$ accordingly. The general training step to mix $k$ images by *TransformMix* is depicted in Algorithm 1.

# 4 Experiments and Results

## 4.1 Experiment Setup

**Mixing Module**. The mixing module comprises a spatial transformer network and a mask prediction network. We follow the implementation of the spatial transformer network (Jaderberg et al., 2015) except the last layer is modified for additional outputs to predict the transformation parameters for two images. The mask prediction network has three convolution layers with 3×3 kernel size, 32 channels and ReLU activation, followed by a 1×1 convolution layer. Padding is applied to ensure that the mixing masks share the same spatial dimension as the input images. The mixing module is trained using SGD with a 0.0005 learning rate and 0.01 weight decay for 100 epochs using a batch size of 128.

**Teacher Network**. In our experiments, the teacher networks are trained on the target datasets following the simple baseline methods. Specifically, the teacher network uses the same network architecture and is trained using the same configurations as the task networks. The test accuracy of the teacher networks on each dataset is the same as the "Simple" baseline listed in Table 1 and 2.

---

**Algorithm 1** Training of *TransformMix*

---

1: **procedure** TRAIN(input images $\{x_i\}_{i=1}^k$, teacher network $f_t$, spatial transformer network $f_s$, mask prediction network $f_m$, task network $g$, mixing parameter $\alpha$)

2:     $S = \{CAM(x_i; f_t)\}_{i=1}^k$

3:     Sample $\Lambda = (\lambda_1, \ldots, \lambda_k)$ from $\text{Dir}(\alpha)$

4:     Sample $z$ from $\text{N}(0, I)$

5:     $\Theta = f_s(S, \Lambda, z)$

6:     $M = f_m(\{\varphi(s_i; \theta_i)\}_{i=1}^k, \Lambda)$

7:     $x' = \sum_{i=1}^k m_i \odot \varphi(x_i; \theta_i)$

8:     $y' = \sum_{i=1}^k \lambda_i y_i$

9:     **if** *SEARCH* **then** *# Training the mixing module*

10:         $\mathcal{L} = CrossEntropyLoss(f_t(x'), y')$

11:         Update $f_s, f_m$

12:     **else** *# Training the task network*

13:         $\mathcal{L} = CrossEntropyLoss(g(x'), y')$

14:         Update $g$

---

## 4.2 Transfer Classification

We study whether a mixing module trained on a dataset can be exploited to create effective mixed data for other datasets. We consider a realistic scenario where a pre-trained classifier is fine-tuned on some downstream tasks. We follow the transfer setting in (Cheung & Yeung, 2022) to fine-tune an ImageNet pre-trained network on four classification datasets: Oxford Flowers (Nilsback & Zisserman, 2008), Oxford-IIIT Pets (Em et al., 2017), FGVC Aircraft (Maji et al., 2013) and Stanford Cars (Krause et al., 2013).

Fine-tuning ImageNet pre-trained model is one of the most promising transfer-learning techniques nowadays. Here, we use ResNet-50 with pre-trained IMAGENET1K_V1 weights provided by `torchvision`[1]. We extract the CAMs of input images from the pre-trained model and fine-tune the model on four downstream datasets: Oxford Flowers (Nilsback & Zisserman, 2008), Oxford-IIIT Pets (Em et al., 2017), FGVC Aircraft (Maji et al., 2013) and Stanford Cars (Krause et al., 2013). Compared to Tiny-ImageNet and ImageNet, these datasets have a larger number of classes but few samples per class. To obtain the CAMs for a new unseen dataset, we use the predictions from the pre-trained teacher network directly as the target class.   FIX

We apply the mixing module pre-trained on TinyImageNet for the TransformMix baseline. For the other baselines, we use the optimal hyperparameters for Tiny-ImageNet suggested by the original papers. The pre-trained model is fine-tuned on the downstream datsets for 100 epochs using a learning rate of 0.001 and batch size of 64. We report the average accuracy and sample standard deviation of our method over five trials. The results in Table 1 demonstrate that *TransformMix* outperforms Mixup, CutMix, SaliencyMix   NEW and PuzzleMix on the Pet, Car and Aircraft datasets and achives similar performance on the Flower datsaet in terms of top-1 and top-5 accuracy.

## 4.3 Direct Classification

Following the line of sample-mixing research, we evaluate *TransformMix* on CIFAR-100 (Krizhevsky & Hinton, 2009), Tiny-ImageNet (Chrabaszcz et al., 2017) and ImageNet (Deng et al., 2009) using WideResNet-28×10 (Zagoruyko & Komodakis, 2016), PreActResNet-18 (He et al., 2016a) and ResNet-50 (He et al., 2016b). We compare our results with simple sample-mixing methods: Mixup (Zhang et al., 2018), Manifold Mixup (Verma et al., 2019), CutMix (Yun et al., 2019) and AugMix (Hendrycks et al., 2020) as well as saliency-aware mixing methods: PuzzleMix (Kim et al., 2020) and SuperMix (Dabouei et al., 2021). In the direct classification experiment, we first train the mixing module (i.e., the spatial transformer network and mask prediction network) on the target dataset, and then utilize the mixing module to create new mixed

---

[1] https://pytorch.org/vision/main/models/generated/torchvision.models.resnet50.html

Table 1: Test-set Top-1 / Top-5 accuracy (%) for fine-tuning a pre-trained network to the Flower, Pet, Car and Aircraft datasets using the mixing module learned from CIFAR-100.

| Method | Flower | Pet | Car | Aircraft |
|---|---|---|---|---|
| Simple | 97.55 / **99.87** | 92.88 / 99.48 | 79.43 / 95.52 | 68.79 / 92.34 |
| Mixup | 97.92 / 99.75 | 93.04 / 99.31 | 80.84 / 95.77 | 70.83 / 93.00 |
| CutMix | 97.67 / 99.63 | 92.09 / 99.20 | 80.91 / 95.94 | 69.63 / 93.06 |
| SaliencyMix | 98.16 / 99.75 | 92.83 / 99.34 | 78.90 / 95.32 | 69.42 / 92.25 |
| PuzzleMix | **98.28** / 99.75 | 92.66 / **99.50** | 80.62 / 95.87 | 70.89 / 92.97 |
| *TransformMix* | 98.06 / 99.71 | **93.32** / **99.51** | **82.27** / **96.27** | **72.33** / **93.47** |
| (ours) | ±0.09 / ±0.05 | ±0.30 / ±0.05 | ±0.21 / ±0.13 | ±0.24 / ±0.39 |

Table 2: Test-set Top-1 / Top-5 accuracy (%) on CIFAR-100, Tiny-ImageNet and ImageNet datasets.

| | CIFAR-100 | | Tiny-ImageNet | ImageNet |
|---|---|---|---|---|
| Method | WRN-28 × 10 | PreActResNet-18 | PreActResNet-18 | ResNet-50 |
| Simple | 78.86 / 93.67 | 76.33 / 91.02 | 57.23 / 73.65 | 75.69 / 92.66 |
| Mixup | 81.73 / 95.02 | 76.84 / 92.42 | 56.59 / 73.02 | 77.01 / 93.52 |
| Manifold MixUp | 82.60 / 95.63 | 79.02 / 93.37 | 58.01 / 74.12 | 76.85 / 93.50 |
| CutMix | 82.50 / 95.31 | 76.80 / 91.91 | 56.67 / 75.52 | 77.08 / 93.45 |
| AugMix | 79.56 / 94.26 | 75.31 / 91.62 | 55.97 / 74.68 | 76.75 / 93.30 |
| PuzzleMix | 84.05 / 96.08 | 80.38 / 94.15 | 63.48 / 81.05 | 77.51 / 93.76 |
| SuperMix | 83.60 / - | - / - | - / - | **77.60** / 93.70 |
| *TransformMix* | **84.07** / **96.97** | **80.39** / **95.37** | **65.72** / **85.01** | **77.60** / **93.89** |
| (ours) | ±0.05 / ±0.07 | ±0.04 / ±0.05 | ±0.16 / ±0.25 | ±0.04 / ±0.06 |

images for training a task classifier from scratch. We repeat the experiments with three random seeds and report the averaged top-1 and top-5 test-set accuracy together with the sample standard deviation in Table 2. FIX

**CIFAR-100**. Following the experiment setting in PuzzleMix (Kim et al., 2020), we train WRN-28×10 (Zagoruyko & Komodakis, 2016) and PreActResNet-18 (He et al., 2016a) on CIFAR-100. We follow the training protocol in (Kim et al., 2020) to train WRN-28×10 for 400 epochs and PreActResNet-18 for 1200 epochs. We use an initial learning rate of 0.1 and decay it by a factor of 0.1 at the $200^{th}$ and $300^{th}$ epochs for WRN-28×10 and at the $400^{th}$ and $800^{th}$ epochs for PreActResNet-18. We adopt the reported baseline performances from (Kim et al., 2020), which are tested under the same experiment setup.

Experimental results show that our method achieves comparable top-1 accuracy as state-of-the-art methods. In addition, *TransformMix* achieves greater gains in the top-5 accuracy with 0.89% and 1.22% improvements over the best performed baseline on WRN-28×10 (Zagoruyko & Komodakis, 2016) and PreActResNet-18 (He et al., 2016a), respectively. This provides evidence that *TransformMix* can preserve more object information, facilitating the task network to learn more effectively.

**Tiny-ImageNet**. Tiny-ImageNet contains 500 training images of 200 classes with a resolution of 64 × 64 (Chrabaszcz et al., 2017). We follow PuzzleMix (Kim et al., 2020) to train the PreActResNet18 network on the Tiny-ImageNet dataset for 1,200 epochs with an initial learning rate of 0.2 and decay it by a factor of 0.1 at the $600^{th}$ and $900^{th}$ epochs. As shown in Table 2, *TransformMix* outperforms PuzzleMix by 2.24% and 3.96% in terms of top-1 and top-5 accuracy, respectively.

**ImageNet**. We also evaluate our methods on the ImageNet dataset with ResNet-50. The dataset contains 1,281,167 training images in 1,000 categories. Following the experiment setup prescribed in (Kim et al., 2020), we train ResNet-50 on the ImageNet for 100 epochs using an initial learning rate of 0.5 with learning rate warm-up and weight decay on resized ImageNet images. Similar to the CIFAR-100 experiment, our method achieves comparable top-1 accuracy and better top-5 accuracy than other baselines (see Table 2).

### 4.4 Object Detection

This section compares *TransformMix* with other mixing baselines on the Pascal VOC and MS-COCO object detection tasks (Everingham et al., 2010). We follow SaliencyMix (Uddin et al., 2021) to use the ResNet-50 pre-trained on ImageNet with different mixing strategies as the backbone network of Faster RCNN (Ren et al., 2015). In the Pascal VOC experiment, we fine-tune the last 5 layers of the backbone networks with the Region Proposal Network and RoI Heads on Pascal VOC 2007 and 2012 data and test on the VOC 2007 test data. Following the same protocol, we fine-tune the model with a batch size of 8 and a learning rate of 0.0004 for 41,000 iterations. We decay the learning rate by a factor of 0.1 at the $33,000^{th}$ iteration. For the COCO dataset, we train and test on the MS-COCO 2017 dataset. We fine-tune the model with a batch size of 8 and a learning rate of 0.02 for 12 epochs. We decay the learning rate by a factor of 0.1 at the $8^{th}$ and $11^{th}$ epochs. As foreground objects are more important than the backgrounds in object detection tasks, saliency-preserving mixing strategies can create more advantageous augmented images to improve detection performance. Specifically, Table 3 shows that *TransformMix* outperforms the simple baseline. Our method also leads other mixing methods in terms of the mAP score on the Pascal VOC dataset and is comparable to other methods on the MS-COCO dataset.

Table 3: Comparison of MixUp, SaliencyMix, PuzzleMix, and *TransformMix* on Pascal VOC object detection task.

|  | Simple | SaliencyMix | PuzzleMix | *TransformMix* |
|---|---|---|---|---|
| Pascal VOC (mAP) | 74.2 | 75.3 | 75.6 | **75.7** |
| MS-COCO (mAP) | 35.9 | **36.5** | **36.5** | **36.5** |

NEW

### 4.5 Execution Time

Similar to SuperMix, *TransformMix* uses a pre-trained teacher network. Although *TransformMix* requires the training of the teacher network and the mixing module during the search phase, *TransformMix* is fast at the inference time. At inference time, our method uses a single forward pass to predict the mixing mask and transformation, while PuzzleMix and SuperMix iteratively compute the optimal mixing masks and transport plans. We test the average processing time to generate a batch of 128 mixed images for 10 trials. All the experiments are conducted using an NVIDIA RTX3090 GPU card. It is found that *TransformMix* is 3.61× faster than PuzzleMix and 3.97× faster than SuperMix on CIFAR-10 and 2.08× faster than PuzzleMix and 18.51× faster than SuperMix on ImageNet (see Table 4).

FIX

Table 4: Comparsion between the execution time of PuzzleMix, SuperMix and *TransformMix*. The execution time of PuzzleMix and SuperMix is represented as the multiply of *TransformMix* execution time.

|  | *TransformMix* | PuzzleMix | SuperMix |
|---|---|---|---|
| CIFAR-10 | 1× | 3.61× | 3.97× |
| ImageNet | 1× | 2.08× | 18.51× |

### 4.6 Ablation Study

In this section, we show the improvements of *TransformMix* over pre-defined transformations and heuristic ways to stack the saliency regions. Additionally, we investigate the effectiveness of different components in *TransformMix*. The details of the ablation baselines are described as follows:

- **Softmax+CAM**: Apply pixel-wise softmax operation to the two extracted CAMs as the mixing masks, and use the mixing masks to combine the two input images.

- **LR**: Move one image to the left and another image to the right. Specifically, we horizontally translate the first image by $\lambda w$ pixels and the second image by $-(1 - \lambda)w$ pixels and stack the two images together. $\lambda$ is the mixing coefficient sample from $\beta(\alpha, \alpha)$ and $w$ is the width of the image.

- **w/o STN**: TransformMix without learned transformations. The mixing mask is computed in the same way as the **Softmax+CAM** baseline.

- **w/o MPN**: TransformMix without learned mixing masks. The two transformed images are stacked together directly.

Table 5: Test-set accuracy (%) on CIFAR-10 and CIFAR-100 with different mixing configurations.

|  | Simple | Mixup | LR | softmax+CAM | w/o MPN | w/o STN | *TransformMix* |
|---|---|---|---|---|---|---|---|
| CIFAR-10 | 95.28 | 95.55 | 95.74 | 95.62 | 96.03 | 96.02 | **96.40** |
| CIFAR-100 | 78.86 | 81.73 | 80.09 | 82.66 | 83.79 | 83.52 | **84.07** |
| TinyImageNet | 57.23 | 57.05 | 56.59 | 63.69 | 63.78 | 64.17 | **65.72** |

The ablation results in Table 5 show that our method is better than the pre-defined left and right translation of images and simple stacking using CAMs as the mixing masks. The results also reveal that the mask prediction network and spatial transformer network contribute to the improvements in *TransformMix*. NEW

### 4.7 Qualitative Analysis

We illustrated the learned mixing strategy in Figure 2 from Section 3. Apparently, the spatial transformer network learns to separate the salient regions of the two input images by squeezing one image to the left and another to the right. Based on the transformed images, the mask prediction network applies the mixing masks that reveal the important areas of the input images. Moreover, the learned temperature value $\tau$ results in a smooth mixing boundary between two objects (see Figure 4). We further validate that *TransformMix* adapts to the mixing coefficient $\lambda$ well in Figure 5. The mask prediction model learns to better reveal the second image (the dog) with the increased value of $\lambda$.

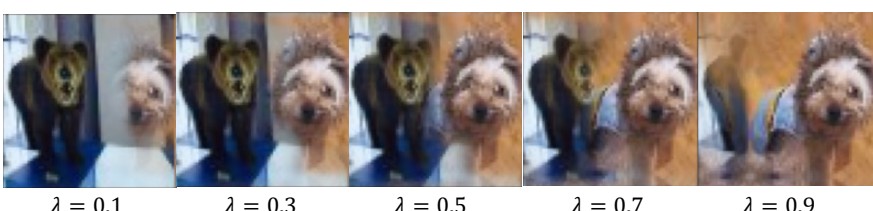

$\lambda = 0.1$ $\quad$ $\lambda = 0.3$ $\quad$ $\lambda = 0.5$ $\quad$ $\lambda = 0.7$ $\quad$ $\lambda = 0.9$

Figure 5: Illustration of the mixed outputs with increasing value of the mixing coefficient $\lambda$.

### 4.8 Mixing More Images

*TransformMix* can be generalized to mix more than 2 images. Figure 6 visualizes the learned mixing strategy when mixing 3 input images on Tiny-ImageNet. The spatial transformer network learns to squeeze the input images to different corners so that the salient regions do not overlap in the mixed results. We compare the end classification performance of *TransformMix* mixing 3 input images on Tiny-ImageNet with Mixup, CutMix and Co-mixup (Kim et al., 2021) in Table 6. Our method achieves higher top-1 and top-5 accuracy scores over other baselines. Using the learned mixing strategy with 3 inputs also achieves slightly better performance than 2 inputs on Tiny-ImageNet.

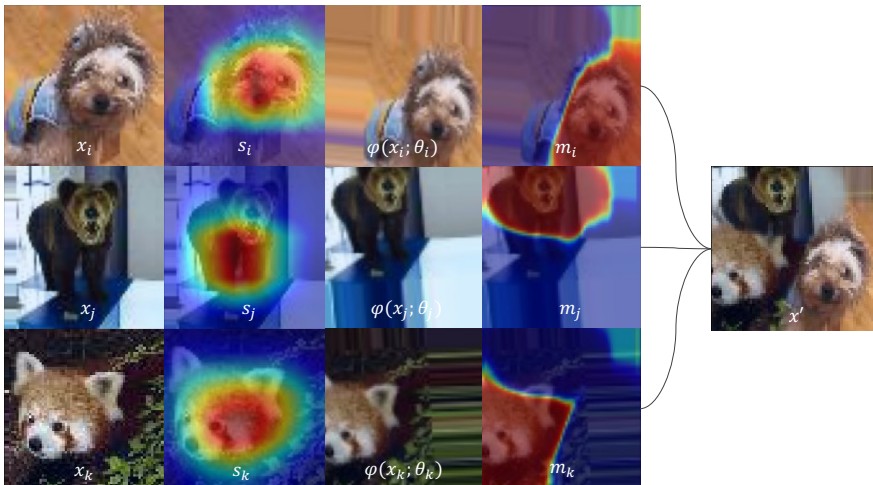

Figure 6: Illustration of the mixing strategy for mixing 3 images from Tiny-ImageNet.

Table 6: The top-1 / top-5 accuracy (%) of different baselines and *TransformMix* with 2 and 3 input images on Tiny-ImageNet.

| Tiny-ImageNet | Simple | Mixup | Cutmix | Co-Mixup | *TransformMix* ($k = 2$) | *TransformMix* ($k = 3$) |
|---|---|---|---|---|---|---|
| top-1 | 57.23 | 56.59 | 56.67 | 64.15 | 65.72 | **66.03** |
| top-5 | 73.65 | 73.02 | 75.52 | - | 85.0 | **85.16** |

## 5 Conclusion

This paper proposes an automated approach, *TransformMix*, to learn transformation and mixing augmentation strategies from data. Based on the class activation maps of the input images, *TransformMix* employs a spatial transformer network to predict the transformation and a mask prediction network to blend the input images. The mixing module is optimized through self-supervision signals given by a pre-trained teacher network efficiently. Through qualitative and quantitive analysis, we demonstrate the effectiveness of *TransformMix* in preserving the salient information and improving the end classification performance on multiple datasets under the direct and transfer settings. As our method does not rely on transformations defined in a specific domain, it is beneficial to study whether the method can be modified and deployed to other domains and data modalities in the future.

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

## A   Appendix

### A.1   Comparing the features between various mixing methods

Table 7: Comparison between the properties of various sample-mixing data augmentation methods.

|  | Mixup | Cutmix | SaliencyMix | PuzzleMix | SuperMix | *TransformMix* |
|---|---|---|---|---|---|---|
| Saliency-aware mixing | ○ | ○ | ● | ● | ● | ● |
| Saliency-aware transformation | ○ | ○ | ◐ | ◐ | ○ | ● |
| Natural blend of images | ○ | ○ | ○ | ○ | ● | ● |
| Effective for image classification | ◐ | ◐ | ◐ | ● | ● | ● |
| Fast inference time | ● | ● | ◐ | ○ | ○ | ◐ |
| Transferable augmentation policy | ○ | ○ | ○ | ○ | ○ | ● |

● = provides property; ◐ = partially provides property; ○ = Not tested or does not provide property

### A.2   Experiment on CIFAR-10

We test the effectiveness of *TransformMix* on the smaller CIFAR-10 dataset and compare the results with Mixup and CutMix. Table 8 shows that *TransformMix* achieves higher top-1 accuracy than the other baselines.

Table 8: Test-set Top-1 accuracy (%) on CIFAR-10 with ResNet-18 and WRN28-10

|  | Simple | Mixup | Cutmix | Transform-Mix |
|---|---|---|---|---|
| ResNet-18 | 95.28 | 95.55 | 96.22 | **96.40**±0.026 |
| WideResNet28-10 | 96.13 | 96.9 | 97.13 | **97.21**±0.016 |

### A.3   Additional experiment on Tiny-ImageNet

We compare the classification accuracy with strong baselines using the training protocol prescribed in Liu et al. (2021). Table 9 shows that our proposed *TransformMix* is a strong method when compared with the AutoMix baseline on Tiny-ImageNet.

### A.4   Comparing Direct and Transfer classification

In this section, we compare the effectiveness between direct TransformMix and transfer TransformMix. Specifically, we utilize the mixing module trained on CIFAR-10 to create new augmented images from unseen CIFAR-100 and TinyImageNet. These new images are used to train a WRN-28×10 model on CIFAR-100 and a PreActResNet-18 model on Tiny-ImageNet from scratch using the same training protocol in the direct classification experiments. Since Tiny-ImageNet has a larger image size than CIFAR-10, we resize the CAMs

Table 9: Test-set Top-1 accuracy (%) for training ResNet-18 network on Tiny-ImageNet.

|  | Simple | MixUp | CutMix | ManifoldMix | SaliencyMix |
|---|---|---|---|---|---|
| Tiny-ImageNet | 61.68 | 63.86 | 65.53 | 64.15 | 64.60 |
|  | PuzzleMix | Co-Mixup | ResizeMix | AutoMix | *TransformMix* |
| (Cont.) | 65.81 | 65.92 | 63.74 | 67.33 | **67.65** |

to match the CIFAR-10 images before inputting them into the spatial transformer network. Table 10 shows that the task models trained with *TransformMix* images outperform those with Input Mixup, Manifold Mixup, and CutMix by a large margin. The experiment results of the transfer mixing module degrade only slightly when compared to the direct mixing module, which is trained on the target datasets.

Table 10: Test-set Top-1 accuracy (%) for training the task network on CIFAR-100 and Tiny-ImageNet datasets using the mixing module learned from CIFAR-10.

|  | Simple | Mixup | Manifold Mixup | CutMix | *TransformMix* (transfer) | *TransformMix* (direct) |
|---|---|---|---|---|---|---|
| CIFAR-100 | 78.86 | 81.73 | 82.60 | 82.50 | $84.02_{\pm0.045}$ | $\mathbf{84.07}_{\pm0.05}$ |
| Tiny-ImageNet | 57.23 | 56.59 | 58.01 | 56.67 | $65.62_{\pm0.29}$ | $\mathbf{65.72}_{\pm0.16}$ |

## A.5 Knowledge Distillation

Table 11 compares *TransformMix* with SuperMix on the knowledge distillation task. We follow the SuperMix experiment, which uses a WRN-40-2 teacher model to provide the supervision signals for training a ShuffleNetV1 student model on the CIFAR-100 dataset with TransformMix augmentation. Despite the efforts to reproduce the results, we could not fully replicate the baseline performance reported by (Dabouei et al., 2021). In our experiments, *TransformMix* improves the accuracy by 7.52% and 0.52% over the student model trained with no sample-mixing and MixUp, respectively. The gains are comparable to the reported results in SuperMix. This demonstrates the effectiveness of *TransformMix* beyond standard classification and object detection tasks.

Table 11: Top-1 test accuracy of the student model trained using knowledge distillation method on CIFAR-100. *Reported results from (Dabouei et al., 2021). †Reproduced results.

|  | SuperMix* | *TransformMix*† |
|---|---|---|
| Teacher Acc. | 75.61 | 75.59 |
| Student Acc. | 70.50 | 69.24 |
| w/ MixUp | 77.44 | 76.24 |
| w/ Method | 78.07 | 76.76 |
| Gain over Student | +7.57 | +7.52 |
| Gain over MixUp | +0.63 | +0.52 |

## A.6 Sensitivity analysis on network configurations

In *TransformMix*, the mask prediction network is implemented as a 3-layer convolutional neural network, where each layer has a channel size of 32 and kernel size of 3x3. We test the sensitivity of our method to the network configurations. Specifically, we compare the end performance of the ResNet-18 models trained on the images created by *TransformMix* with different implementations of the mask prediction network. The task model is trained on the CIFAR-100 dataset for 200 epochs, and the mask prediction network uses a different

number of layers, channel size, and kernel size from $\{2, 3, 4\}$, $\{8, 16, 32, 64\}$ and $\{3, 5, 7\}$, respectively. We also explore the effect of using different $\alpha$ values in $\{0.2, 0.5, 1, 2, 3, 4\}$ when sampling the mixing coefficients. The experimental results illustrated in Figure 7a, 7b, 7c and 7d show that *TransformMix* is not sensitive to the neural architecture design and the $\alpha$ parameter. It is also worth noting that our method does not require fine-tuning additional parameters on new datasets. This is an advantage over existing methods that require manual specification of additional hyperparameters, such as the label smoothness, data local smoothness, transport cost, and neighbor size in PuzzleMix Kim et al. (2020); the roughness and sparsity coefficients in SuperMix Dabouei et al. (2021).

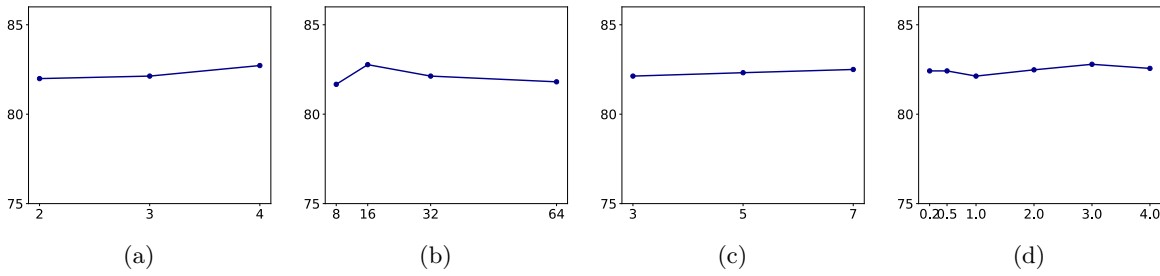

Figure 7: Sensitivity to different choices of (a) number of layers, (b) number of channels, (c) kernel size of the mask prediction network and (d) mixing parameter $\alpha$. The vertical axis shows the top-1 test-set accuracy (%) and the horizontal axis shows the values of the tested configuration.

## A.7 More visual illustrations of mixed output generated by TransformMix

We present more visualizations of the intermediate results and mixed images generated by *TransformMix* on different datasets in Figure 8 and Figure 9. The illustrations show that *TransformMix* learns to separate the salient regions of the input images and apply appropriate mixing masks to expose the salient regions on the mixed results.

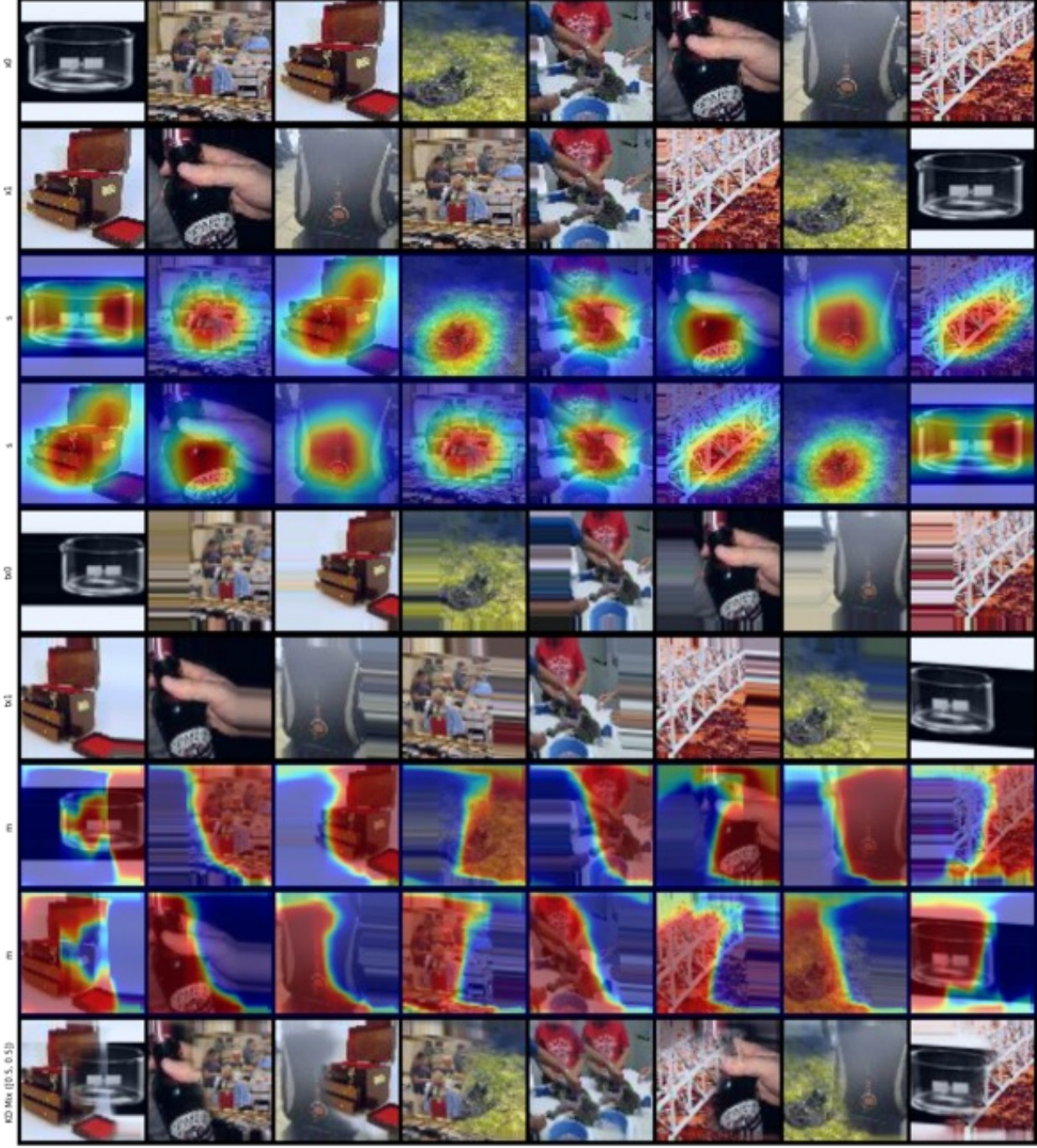

Figure 8: More illustration of the mixed outputs from a non-curated set of eight images. From top to bottom, the row indicates input image 1, input image 2, CAM of input image 1, input image 2, transformed input image 1, transformed input image 2, predicted mask of input image 1, the predicted mask of input image 2, and mixed result.

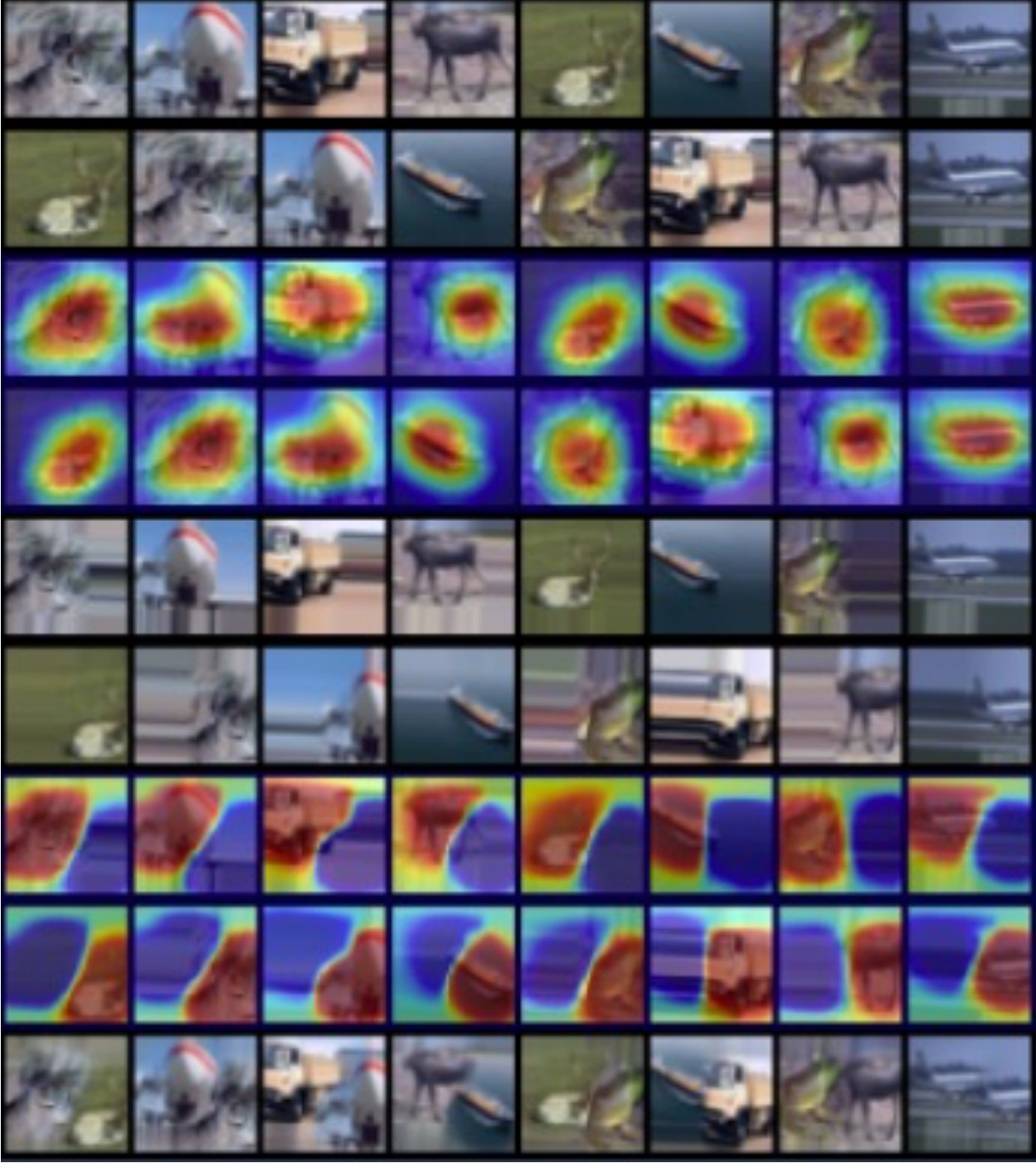

Figure 9: More illustration of the mixed outputs from a non-curated set of eight images. From top to bottom, the row indicates input image 1, input image 2, CAM of input image 1, input image 2, transformed input image 1, transformed input image 2, predicted mask of input image 1, the predicted mask of input image 2, and mixed result.

