# OpenReview forum: "TransformMix: Learning Transformation and Mixing Strategies from Data"
_TMLR — Rejected by TMLR_

### Review · Reviewer_TTvK · 2024-04-11

**Summary Of Contributions:**

This work introduces a new learnable data augmentation strategy, TransformMix. TransformMix is based on image mixing augmentation, which combines two or few images and their corresponding labels to produce more training data. Given the class activation maps (CAM) produced by a teacher model, a Spatial Transformer Network (STN) generates affine transformation parameters for the images, and then a Mask Prediction Network (MPN) predicts masks for mixing the transformed images into one. TransformMix shows competitive performance in various settings and tasks.

**Audience:**

Yes

**Broader Impact Concerns:**

The reviewer does not have concerns regarding broader impacts of this work.

**Claims And Evidence:**

Yes

**Requested Changes:**

Please check the weaknesses.

**Strengths And Weaknesses:**

Strengths
1. The learnable data augmentation strategy is able to capture semantically significant regions in the image and take such regions into account when mixing multiple images. Compared with prior methods which do not examine the image contents while mixing them, TransformMix can produce more reliable augmented training data.

2. The modules are lightweight and require less execution time than previous methods like PuzzleMix and SuperMix.

3. TransformMix demonstrates strong transferable performance in classification and object detection tasks.

Weaknesses
1. The experiments about transfer learning for object detection seem to be a bit out-dated (Faster R-CNN on Pascal VOC 2012). It would be better to evaluate with more modern and widely used datasets (e.g., MS-COCO and LVIS). Instance/Semantic/Panoptic segmentation tasks may also be considered for assessing model transferability.

2. From Figure 2, 8, and 9, STN always places one object on the left and another on the right, with roughly equal sizes. There does not seem to be much diversity in layouts. As a result, there may be a domain gap between such synthesized images and real-world images captured from complex scenes. It is concerning whether such a bias/gap would negatively affect the final performance.

3. Before applicable to the task network, TransformMix requires first pre-training a teacher model, generating CAMs from the teacher, and learning the STN+MPN with the teacher and CAMs. The computation is indeed reduced at inference time, but when comparing efficiency, such overheads should also be taken into consideration. It is recommended to include training costs into the comparison.

---

> ### Author Response · Authors · 2024-04-27
> **Thanks for the review. We follow the requested changes to add new object detection experiments, clarify the domain gap, and acknowledge the additional training effort**
>
> __COCO dataset.__ We add the COCO experiment in Section 4.4. Using TransformMix improves the simple baseline that does not use any mixing strategy. The improvement is comparable with other saliency-aware mixing methods. We understand that the results may not beat the state-of-the-art object detection benchmark, our major focus is to show that TransformMix can be deployed to different computer vision tasks beyond classification.
>
> __Domain gap between synthesized images and real-world images.__
> We agree that there may be a domain gap between synthesized images and real-world images in complex scenes. However, this does not hinder the improvement in classification capability through training on mixed images. For example, MixUp creates unrealistic interpolated images, but the method consistently improves classification baselines across multiple tasks. We would also like to mention that even if the domain gap exists, our method outputs more visually appealing results compared to other methods (without unnatural boundaries and artifacts), as shown in Fig. 1. We hope this can mitigate the concern about the discrepancy between mixed samples and real-world data.
>
> __Training Time.__ Thanks for pointing this out. We acknowledge the additional training time during the search phase in Section 4.5.

---

### Review · Reviewer_wirf · 2024-04-15

**Summary Of Contributions:**

This manuscript proposed a new mixing strategies for vision tasks that leveraging a pre-trained teacher network to detect the saliency region of different visual input and then combine the saliency region to form the augmented data. The authors conduct lots of the experiments to justify the idea.

**Audience:**

Yes

**Broader Impact Concerns:**

N/A.

**Claims And Evidence:**

Yes

**Requested Changes:**

I would rather ask to perform more ablation on the proposed methods. Specifically, the authors propose the methods based on several assumptions, e.g. learned transformation can help more compared with pre-defined transformation. I would like to see more experiments demonstrating these assumptions.

**Strengths And Weaknesses:**

## Strengths:
* The idea is simple and easy to follow.

## Weaknesses:
* I feel the motivation of the proposed method is not that clear. For example, if the core idea is to use the saliency region, what’s the benefit of the proposed method compared with just extracting the saliency region and do some stacking to generate the new sample?

---

> ### Author Response · Authors · 2024-04-27
> **Thanks for the review. We follow the requested changes to add more ablation studies and clarify our motivation.**
>
> __Motivation.__ Thanks for your question. From previous works, we observe that the mixing strategy is mostly designed manually, for example, by interpolating images (MixUp), randomly pasting an image patch to another image (CutMix), or pasting the saliency region from an image to another image (SaliencyMix). We believe there is an automated way to learn a mixing strategy from the data instead of defining how two inputs should be combined. It may be easy to define the mixing strategy manually for images. However, this may not be the case for other data modalities. Therefore, we believe studying an automated framework that automatically learns the mixing strategy is beneficial to the development of smarter mixing strategies in other domains. This motivates us to propose an automated framework that learns good transformation and mixing masks automatically.
>
> Compared with simple stacking and pre-defined transformation. These operations are usually applied regardless of the image content. This may cause the saliency regions to be overlapped or diluted during the mixing process. With our proposed learning framework, the mixing modules learn to create good mixed results. These mixed results are shown to be more effective than simple mixing and previous saliency-aware mixing methods.
>
> __More Ablation Studies.__ In order to support our claims, we add additional ablation studies to show that the learned transformation and mixing masks are more effective than pre-defined transformation and simple stacking of saliency regions. We created four ablation baselines: LR, Softmax+CAM, w/o STN, and w/o MPN to verify our claims. The description of these baselines is as follows:
>
> - __LR__: Moving one image to the left and another image to the right. Specifically, we horizontally translate the first image by $\lambda w$ pixels and the second image by $-(1-\lambda)w$ pixels and stack the two images together. $\lambda$ is the mixing coefficient sampled from $\beta(\alpha, \alpha)$ and $w$ is the width of the images.
> - __Softmax+CAM__: Apply pixel-wise softmax operation to the two extracted CAMs as the mixing masks, and use the mixing masks to combine the two input images.
> - __w/o STN__: TransformMix without learned transformations.
> - __w/o MPN__: TransformMix without learned mixing masks.
>
> |  | Simple | MixUp | LR | softmax+CAM | w/o MPN | w/o STN | ours |
> | --- | --- | --- | --- | --- | --- | --- | --- |
> | CIFAR-10  	| 95.28 | 95.55  | 95.74 | 95.62 		| 96.03		 | 96.02 | __96.40__ |
> | CIFAR-100	 | 78.86 |  81.73|  80.09| 82.66 	| 83.79 		| 83.52 | __84.07__ |
> | TinyImageNet | 57.23 | 57.05 | 56.59 | 63.69 		| 63.78 		| 64.17 |  __65.72__ |
> _Table 6. Ablation Studies._
>
> The ablation results in Table 6 show that our method is better than the pre-defined left and right translation of images (LR) and simple stacking using CAMs as the mixing masks (softmax+CAM). We further validate the effectiveness of learned transformations and learned mixing mask prediction modules as shown in the w/o MPN and w/o STN baseline. We hope that these studies can address the concerns about the effectiveness of our method.

---

### Review · Reviewer_t8ib · 2024-04-18

**Summary Of Contributions:**

The work considers training a saliency-based mixup strategy, specific to image classification networks. The mixing module trains both a transformation (via a spatial transformer network, STN) and a masking map for combining images, based on class activation maps (CAM) obtained from a trained version (without any sort of data augmentation or mixup) of the initial network. They also propose a transfer setup, where the mixing module is trained on one dataset, and then the model is fine-tuned on a different dataset.

**Audience:**

Yes

**Broader Impact Concerns:**

The work does not have broader ethical concerns beyond those of any general image-based deep learning works, and I do not think it requires a broader impact statement.

**Claims And Evidence:**

Yes

**Requested Changes:**

I have a few requests that would be nice to have:
* The additional time for training of the teacher network and mixing module should be acknowledged in the text, if not included explicitly in the results section.
* For all experiments, the number of runs should be reported, and sample standard deviation (1/(N-1) factor) should be used (and usage of it as opposed to population stdev should be noted). Ideally, more runs would be best to avoid misleading error bar estimates, but I understand if computing resources are a challenge.
* There are many Mixup variants out there that were missed and would be nice to reference at least:
   * Comixup (Kim et al. 21); would also be a good one to compare to given that it's a later paper from the same authors of PuzzleMix
   * K-Mixup (Greenewald et al. 23); a simple generalization of Mixup
   * StackMix (Chen et al. 22)
   * GAN-Mixup (Sohn et al. 21)
   * Local Mixup (Baena et al. 24)
* Clarity could be improved in two spots, in my opinion:
   * For the use of the STN, it would be best to also note that the 6 parameters are just those of an affine transformation on the image (2 x 2 for linear part, plus 2 for translation).
   * Clarify what is done in the transfer learning experiments (see note above).

**Strengths And Weaknesses:**

Strengths:
* The method qualitatively seems to produce output that is more appropriately mixed than competing methods.
* Experiments seem to show small, but consistent improvement over existing methods (a bit of concern on error bars, see below).
* The ablation study to show necessity of different parts of the framework is certainly appreciated.

Weaknesses:
* The method requires pre-training of a teacher network, then subsequent training of a mixing module, before the final training of the model itself. The first two steps are certainly nontrivial in time and computing resource investment. The execution time experiments only show the time required to generate mixed data instances after these first two steps.
* The method does not generalize to other data types, text, audio, etc, which is true of the vanilla Mixup for example.
* It is unclear how reliable the reported standard deviations are. In particular, in the only spot where they mention the number of instances ran (CIFAR-100 in Sec. 4.3), the authors state that only three instances are run, if I understood correctly. The sample standard deviation is heavily biased downward in such a small sample size, so the reported error bars may be a little misleading.
* I found the transfer setup to be rather confusing, to be honest. How does one obtain the CAMs if there are different classes for the other datasets? If this teacher network is fine-tuned with new classes, and thus the new dataset, this does not seem to be a true instance of transfer learning, as the mixing module also has some interaction with the downstream datasets via the teacher network.
* It is unclear to me how much the efficacy of the teacher network affects the overall method. Would using Mixup, or yet another round of this procedure for training the teacher model used in training the mixing module give further gains, or are there diminishing returns at this point?

---

> ### Author Response · Authors · 2024-04-27
> **Thanks for the valuable suggestions. We followed the requested changes and updated our paper.**
>
> __Training Time.__ We acknowledged the additional training time during the search phase in Section 4.5.
>
> __Mixing other types of data.__ Similar to other saliency-aware methods, like PuzzleMix and SuperMix, our method cannot directly transfer to other data modalities. However, compared with other works, our method provides an __automated framework__, i.e., utilizing a teacher network and introducing a mixing module to create mixed results automatically. If an appropriate mixing module is available in any new data modality, we believe our automated framework can be applied to the new data type.
>
> __Number of experiment runs.__ Thanks for pointing this out. We should explicitly state that we ran all experiments for three trials and reported the sample standard deviation. In the new revision, we increase the number of trials from 3 to 5 for our major Transfer Classification experiments in Section 4.2. We also state the number of trials and the use of sample standard deviation in Sections 4.2 and 4.3.
>
> __Transfer Setting.__ We apologize for not clearly mentioning the way to extract the CAMs in the transfer setting. We would like to clarify the transfer setting. To obtain the CAMs for a new unseen dataset, we use the predictions from the pre-trained teacher network directly as the target class. The teacher network is not fine-tuned on the target dataset, and it does not violate the transfer setting assumption.
>
> __More MixUp references.__ Thanks for the information. We added the discussion of the mentioned MixUp variants to the related section in the latest version of our manuscript.
>
> __STN parameters.__ It is a good idea to clarify the 6 affine transformation parameters. In the Spatial Transfer Network section under Section 3.1, we added the explanation of the six parameters,  a $2\times2$ sub-matrix that defines the linear transformation, and 2 parameters that define the horizontal and vertical translations.
>
> __Efficacy of the teacher network.__ It is interesting to study how the efficacy of the teacher network affects the learning of the mixing and transformation strategies. As suggested, we tested using a network trained with MixUp data as the teacher network. We found that the results are mostly the same as the original setting. The transformations and mixing masks that were learned are similar: the input images are transformed and merged so that the saliency regions are preserved in the mixed results.

---

### Author Response · Authors · 2024-04-27
**Summary of Changes.**

We thank all the reviewers for their valuable and quality reviews. In the current version, we added new changes to clarify our methods and new content to support our claims. The changes are colored in violet in the new revision. The clarifications are marked with the ‘FIX’ tags, and the new content is marked with the ‘NEW’ tags.
The major changes are as follows:

[FIX]
1. We clarified the six affine parameters of the spatial transformation network in Section 3.1.
2. We explained how to use pre-trained teacher networks to obtain CAMs for unseen datasets in Section 4.2.
3. We stated the number of experimental trials and the use of sample standard deviation when reporting the results in Transfer Classification and Direct Classification in Sections 4.2 and 4.3.
4. We acknowledged the extra time for training the teacher network and mixing module in Section 4.5.

[NEW]
1. We added more references to five MixUp variants, as suggested by one of our reviewers.
2. We ran more trials (from 3 to 5) on our major results in the transfer classification experiments.
3. We added the experiment on the MS-COCO dataset in Section 4.4.
4. In Section 4.6, we added new Ablation studies to show the effectiveness of our method compared to simple stacking and pre-defined transformations.

---

### Decision · Action_Editor_GoKq · 2024-05-30

**Recommendation:** Reject

**Comment:**

After the authors’ revision, the remaining concerns from the reviewers primarily focus on the effectiveness of the proposed data augmentation strategy for object detection on the MS-COCO dataset. The performance of the proposed method does not show improvement over existing baselines and appears to be inferior to the plain baseline with standard training recipes. Since object detection and further semantic segmentation are crucial tasks for demonstrating the usefulness of the proposed data augmentation strategy, performance on this testbed is critical.

In addition, while the reviewers acknowledge the speed-up of the proposed method during inference, they noted an increased overhead during training. The reviewers felt that this point was not made clear in either the original submission or the revision. Specifically, Reviewer TTvK pointed out that “the revision only ‘acknowledges’ the additional training time but does not directly show the actual overhead.”

Furthermore, the reviewer noted an inherent bias in the proposed method toward generating left-and-right layouts. Properly addressing this bias could potentially strengthen the method.

Based on these points, I recommend rejection to encourage the authors to further improve their method. They need to convincingly demonstrate performance gains in object detection tasks and show that these gains justify the training overhead.

**Audience:**

Researchers and practitioners working on image recognition and object detection might be interested in reading this paper.

**Claims And Evidence:**

Summary:

This paper proposes TransformMix, a novel data augmentation strategy that produces additional training data by learning to mix up existing images and their corresponding labels. The key idea involves using a teacher model, a saliency detector, and a mixing module to guide the process. Specifically, the class activation maps (CAM) produced by the teacher model inform a Spatial Transformer Network (STN), which generates affine transformation parameters for the images. Following this, a Mask Prediction Network (MPN) predicts masks for merging the transformed images into a single composite. Evaluations on several image classification and object detection tasks demonstrate the effectiveness of the proposed TransformMix.

Claims:

The key claims made in the paper are that the proposed TransformMix strategy (1) is an automatically learned data augmentation method that explicitly preserves the visual saliency of input images; (2) can be transferred to augment unseen datasets; and (3) offers advantages over existing sample-mixing methods, proving effective across various tasks including classification, object detection, and knowledge distillation.

Evidence:

The evidence provided in the paper mostly supports the claims, but evidence for Claim 3 is a bit weak. As Reviewer TTvK pointed out, “in the revision, additional results of training a Faster R-CNN/ResNet-50 on MS-COCO are shown. The proposed method, TransformMix, is on par with other baselines like SaliencyMix (36.5 mAP). The results are not strong enough to support the proposed data augmentation strategy. In fact, a Faster R-CNN/ResNet-50 can reach 37+ mAP with standard training recipes.” In addition, reviewers (TTvK, t8ib) pointed out that the proposed method “requires quite a bit of overhead with the teacher network and mixing module, which are certainly nontrivial in time and computing resource investment.”

For Claim 2 regarding the generalizability of the proposed method, Sec. A.4 provides supportive evidence by demonstrating that the mixing module, trained on CIFAR-10, can create new augmented images from unseen datasets like CIFAR-100 and TinyImageNet. However, Reviewer t8ib noted that “the method does not generalize to other data types, such as text or audio.”

Claim 1 is well supported by the characteristics of the proposed method, along with the ablation studies presented in both the original submission and the revision.

**Resubmission Of Major Revision:**

The authors may consider submitting a major revision at a later time.